

# Explaining neural networks for detection of tropical cyclones and atmospheric rivers in gridded atmospheric simulation data

Tim Radke[1], Susanne Fuchs[1], Christian Wilms[3], Iuliia Polkova[2,4,5], and Marc Rautenhaus[1,2]

[1]Visual Data Analysis Group, Universität Hamburg, Hamburg, 20146, Germany

[2]Center for Earth System Research and Sustainability (CEN), Universität Hamburg, Hamburg, 20146, Germany

[3]Computer Vision Group, Universität Hamburg, Hamburg, 22527, Germany

[4]Institute of Oceanography, Universität Hamburg, Hamburg, 20146, Germany

[5] now at Deutscher Wetterdienst, Offenbach am Main, 63067, Germany

*Correspondence to*: Tim Radke (tim.radke@uni-hamburg.de)

**Abstract.** Detection of atmospheric features in gridded datasets from numerical simulation models is typically done by means of rule-based algorithms. Recently, also the feasibility of learning feature detection tasks using supervised learning with convolutional neural networks (CNNs) has been demonstrated. This approach corresponds to semantic segmentation tasks widely investigated in computer vision. However, while in recent studies the performance of CNNs was shown to be comparable to human experts, CNNs are largely treated as a "black box", and it remains unclear whether they learn the features for the correct reasons. Here we build on the recently published "ClimateNet" dataset that contains features of tropical cyclones and atmospheric rivers as detected by human experts. We adapt the explainable artificial intelligence technique "Layer-wise Relevance Propagation" (LRP) to the feature detection task and investigate which input information CNNs with the Context-Guided Network (CG-Net) and U-Net architectures use for feature detection. We find that both CNNs indeed consider plausible patterns in the input fields of atmospheric variables, which helps to build trust in the approach. We also demonstrate application of the approach for finding the most relevant input variables and evaluating detection robustness when changing the input domain. However, LRP in its current form cannot explain shape information used by the CNNs, and care needs to be taken regarding the normalization of input values, as LRP cannot explain the contribution of bias neurons, accounting for inputs close to zero. These shortcomings need to be addressed by future work to obtain a more complete explanation of CNNs for geoscientific feature detection.



## 1. Introduction

The automated detection and tracking of 2-D and 3-D atmospheric features including cyclones, fronts, jet streams, or atmospheric rivers (ARs) in simulation and observation data has multiple applications in meteorology. For example, automatically detected features are used for weather forecasting (e.g., Hewson and Titley, 2010; Mittermaier et al., 2016;

Hengstebeck et al., 2018), statistical and climatological studies (e.g., Dawe and Austin, 2012, Pena-Ortiz et al., 2013, Schemm et al., 2015, Sprenger et al., 2017, Lawrence and Manney, 2018), and visual data analysis (e.g., Rautenhaus et al., 2018; Bösiger et al., 2022; Beckert et al 2023). Features are typically objectively detected based on a set of physical and mathematical rules. For example, cyclones can be identified by means of searching for minima or maxima in variables including mean sea level pressure and lower-tropospheric vorticity (Neu et al., 2013; Bourdin et al. 2022), atmospheric

fronts by means of derivatives of a thermal variable combined with threshold-based filters (Jenkner et al., 2010; Hewson and Titley, 2010; Beckert et al., 2023), and ARs based on thresholding and geometric requirements (Guan and Waliser, 2015; Shields et al., 2018).

Recent research, however, has shown that given a pre-defined labelled dataset, supervised learning with artificial neural networks (ANNs), in particular convolutional neural networks (CNNs), can learn a feature detection task. For example,

Kapp-Schwoerer et al. (2020) and Prabhat et al. (2021) (abbreviated as KS20 and P21 hereafter) showed that CNNs can be trained to detect tropical cyclone (TC) and AR features, and Lagerquist et al. (2019), Biard and Kunkel (2019), Niebler et al. (2022), and Justin et al. (2023) used CNNs to detect atmospheric fronts. In these works, CNNs are used to classify individual grid points of a gridded input dataset according to whether they belong to a feature. This corresponds to a "semantic segmentation task" widely investigated in the computer vision literature for segmentation and classification of regions in

digital images, e.g., cars, trees, or road surface (Long et al., 2015; Liu et al., 2019; Xie et al., 2021).

Using CNNs for feature detection via semantic segmentation can have several advantages, including increased computational performance (Boukabara et al., 2021; Higgins et al., 2023) and the option to learn features that are difficult to formulate as a set of physical rules (P21; Niebler et al., 2022; Tian et al., 2023). A major limiting factor, however, is that they are "black box" algorithms that do not allow for an easy interpretation of the decision-making process inside CNNs.

Hence, one does not know whether a CNN bases its decision on the correct patterns in the data. If not, the CNN may still perform well on the training data but fails to generalize to unseen data (Lapuschkin et al., 2019). To approach this issue, the artificial intelligence (AI) community has proposed methods for explainable artificial intelligence (xAI) in the past decade (Linardatos et al., 2021; Holzinger et al., 2022), including Layer-wise Relevance Propagation (LRP; Bach et al., 2015), Local Interpretable Model-Agnostic Explanations (LIME; Ribeiro et al., 2016), Gradient-weighted Class Activation

Mapping (Grad-CAM; Selvaraju et al., 2017), and Shapley Additive Explanations (SHAP; Lundberg and Lee, 2017). In short, these methods provide information about what an ANN "looks at" when computing its output, hence allow evaluation of the plausibility of the learned patterns.



The xAI methods vary with respect to several characteristics, including whether the relevance of the input data is computed per input grid point[1] or for entire regions of the input data and per input variable[2], or jointly for all variables. Also, the complexity of implementation differs. For application in semantic segmentation, an open challenge is also that existing xAI methods have been mainly developed for classification tasks, i.e., for CNNs assigning input data to one of several classes (this corresponds to the question "is a particular feature contained in the input data" instead of identifying the spatial structure of a feature). While Grad-CAM is readily available for use with semantic segmentation (Captum, 2023; MathWorks, 2023), it has the drawback of not being able to differentiate between input variables (Selvaraju et al., 2017). SHAP can be implemented for semantic segmentation (e.g., Dardouillet et al., 2023), however, it computes relevance values for clusters of input grid points and not for individual input grid points. The same applies to LIME; moreover, to the best of our knowledge, we are not aware of implementations of LIME for semantic segmentation. The feasibility of using LRP for semantic segmentation has been demonstrated in the context of medical imaging (Tjoa et al., 2019; M. A. Ahmed and Ali, 2021) and it produces relevance information per grid point and input variable.

Our goal for the study at hand is to provide an xAI method that works with semantic segmentation CNNs trained to detect atmospheric features. We are interested in opening the "black box" to investigate whether a CNN uses physically plausible input patterns to make its decision. This requires analysis of the spatial distribution of input relevance (i.e., which regions and structures are relevant for a particular feature; hence relevance information *per grid point* is needed), as well as analysis of distributions of relevant input variables (i.e., which values of which input variables are of importance to detect a feature; hence relevance information *per input variable* are needed).

As application example, we consider the work by KS20 and P21, who introduced an expert-labelled dataset of TCs and ARs in atmospheric simulation data (the "ClimateNet" dataset), and trained two different CNN architectures, DeepLabv3+ (Chen et al., 2018) and Context-Guided Network (CG-Net; Wu et al., 2021), to perform feature detection via semantic segmentation. The studies showed that for the given task, the CNNs learnt to detect TCs and ARs and that the CG-Net architecture outperformed the DeepLabv3+ architecture. However, in neither study an xAI technique was applied.

Mamalakis et al. (2022) (abbreviated as M22 hereafter) recently presented work in this direction by reformulating the P21 segmentation task into a classification task and evaluating several available xAI techniques for classification, including LRP and SHAP. They considered sub-regions of the global dataset used by P21 and differentiated whether zero, one, or more ARs exist in a sub-region. TCs were not considered. M22 showed that for their classification setup, LRP yielded useful information to assess the plausibility of the decision-making inside the CNN. LRP has also been successfully applied in further geoscientific studies concerned with use of CNNs for classification tasks (Toms et al., 2020; Davenport and

---

[1] Computer vision literature concerned with image data uses the term "pixel" for individual input data points. In this study we are concerned with gridded simulation data and hence use the term "grid point".

[2] Similarly, we use the term "input variable" instead of "colour channel" commonly used in computer vision.



Diffenbaugh, 2021; Labe and Barnes, 2022; Beobide-Arsuaga et al., 2023), and it fulfils our requirement of computing relevance information per grid point and input variable (at least in some variants; cf. M22).

In this study, we build on the work by KS20, P21, and M22. We demonstrate and analyse the use of LRP for the KS20/P21 case of detecting TCs and ARs, using the CG-Net architecture used by KS20 and the ClimateNet dataset provided by P21 (Sect. 2). We reproduce the KS20/P21 setup (Sect. 3) and address the following objectives:

1. Adapt LRP to the semantic segmentation task for geoscientific datasets and extend the method to be applicable to the CG-Net CNN architecture (Sect. 4).

2. Examine the plausibility of spatial relevance patterns and distributions of relevant inputs for TC and AR detection as computed with LRP (Sects. 5 and 6).

3. Demonstrate further applications of LRP for semantic segmentation, including assessment of the most relevant input variables for a feature detection task and assessment of the robustness of feature detection when data of sub-regions instead of global data is used as input (Sect. 7).

For comparison and due to its widespread use for semantic segmentation in computer vision, we also consider the U-Net architecture (Ronneberger et al., 2015). To limit paper length, however, its results are mainly presented in the Supplement.

## 2. The ClimateNet dataset

The ClimateNet dataset introduced by P21 contains global 2-D longitude-latitude grids of selected atmospheric variables at a collection of time steps simulated with the Community Atmospheric Model (CAM5.1), spanning a time interval from 1996 to 2013. Each grid has a size of 768×1152 grid points and contains 16 variables, listed in Table 1. Experts labelled 219 time steps, assigning each grid point to one of three classes: background (BG), TC, and AR. An individual feature is represented by connected grid points of the same class. As most time steps were labelled by multiple experts, the dataset contains 459 input-output mappings, with sometimes very different labels for the same input data. As P21 argued, these disagreements in classifications reflect the diversity in views and assumptions by different experts. P21 split the labelled data into a training (398 mappings) and test dataset (61 mappings) by taking all time steps prior to 2011 as training data and all other as test data.

Following KS20, we apply z-score normalization on each variable to set the mean values to 0 and the standard deviations to 1, hence achieving equally distributed inputs. As discussed by LeCun et al. (2012), this normalization reduces the convergence time of CNNs during training. Also, z-score normalization helps to treat all input variables equally important by a CNN (e.g., Chase et al., 2022). An issue when using LRP (and other xAI methods; cf. M22) with z-score normalized data, however, is the "ignorant-to-zero-input issue" discussed by M22: zero input values are assigned zero relevance. We will discuss the impact of this issue on the usefulness of the LRP results. For comparison, we also discuss results obtained by training the CNNs using a min-max normalization (which rescales the variable values to the range [0, 1]; e.g., García et al.,





2014) and a modified z-score normalization shifted by a value of +10 in the normalised data domain (the mean value becomes +10; the standard deviation remains 1).


**Table 1: Atmospheric 2-D fields (variables) contained the P21 ClimateNet dataset. Variable short names are used throughout the text.**

| Variable | Description | Mean | Standard Dev. | Units |
|---|---|---|---|---|
| U850 | Zonal wind at 850 mbar pressure surface | 1.56 | 8.29 | m/s |
| V850 | Meridional wind at 850 mbar pressure surface | 0.270 | 6.22 | m/s |
| UBOT | Lowest level zonal wind | 0.129 | 6.65 | m/s |
| VBOT | Lowest model level meridional wind | 0.332 | 5.77 | m/s |
| TS | Surface temperature (radiative) | 271 | 23.7 | K |
| T200 | Temperature at 200 mbar pressure surface | 213 | 7.99 | K |
| T500 | Temperature at 500 mbar pressure surface | 253 | 12.8 | K |
| TREFHT | Reference height temperature | 279 | 22.5 | K |
| TMQ | Total (vertically integrated) precipitable water | 19.3 | 15.8 | $kg/m^2$ |
| QREFHT | Reference height humidity | $7.83 \times 10^{-3}$ | $6.20 \times 10^{-3}$ | kg/kg |
| PRECT | Total (convective and large-scale) precipitation rate (liq + ice) | $2.95 \times 10^{-8}$ | $1.56 \times 10^{-7}$ | m/s |
| ZBOT | Lowest model level height | 61.3 | 4.91 | m |
| Z200 | Geopotential Z at 200 mbar pressure surface | $11.7 \times 10^3$ | $0.635 \times 10^3$ | m |
| Z1000 | Geopotential Z at 1000 mbar pressure surface | 474 | 833 | m |
| PS | Surface pressure | $96.6 \times 10^3$ | $9.71 \times 10^3$ | Pa |
| PSL | Sea level pressure | $101 \times 10^3$ | $1.46 \times 10^3$ | Pa |

## 3. Reproduction of the KS20/P21 task with CG-Net and U-Net

Following KS20/P21, we formulate the detection of TCs and ARs as a semantic segmentation task, with the goal of
assigning one of the classes TC, AR, or BG to every grid point. We evaluate the CG-Net (Wu et al., 2021; shown by KS20 to outperform the DeepLabv3+ architecture used by P21) and U-Net (Ronneberger et al., 2015) CNN architectures. Figure 1 illustrates both architectures. CNNs are a class of ANNs that capture spatial patterns by successively convolving the data with spatially local kernels (e.g., Russell and Norvig, 2021). For semantic segmentation tasks, CNNs compute as output a probability value for each grid point and class. The U-Net features the characteristic encoder-decoder architecture that first
successively decreases the grid size to detect high-level patterns at different scales using convolutional layers, followed by



upsampling and a combination of the extracted patterns leading to segmentation as output. To improve the quality of the segmentation, skip connections between the respective levels of the encoder and decoder are introduced. In contrast, CG-Net uses a typical classification-style CNN without a dedicated decoder and introduces context guided blocks that combine spatially local patterns with larger-scale patterns to produce a final segmentation.


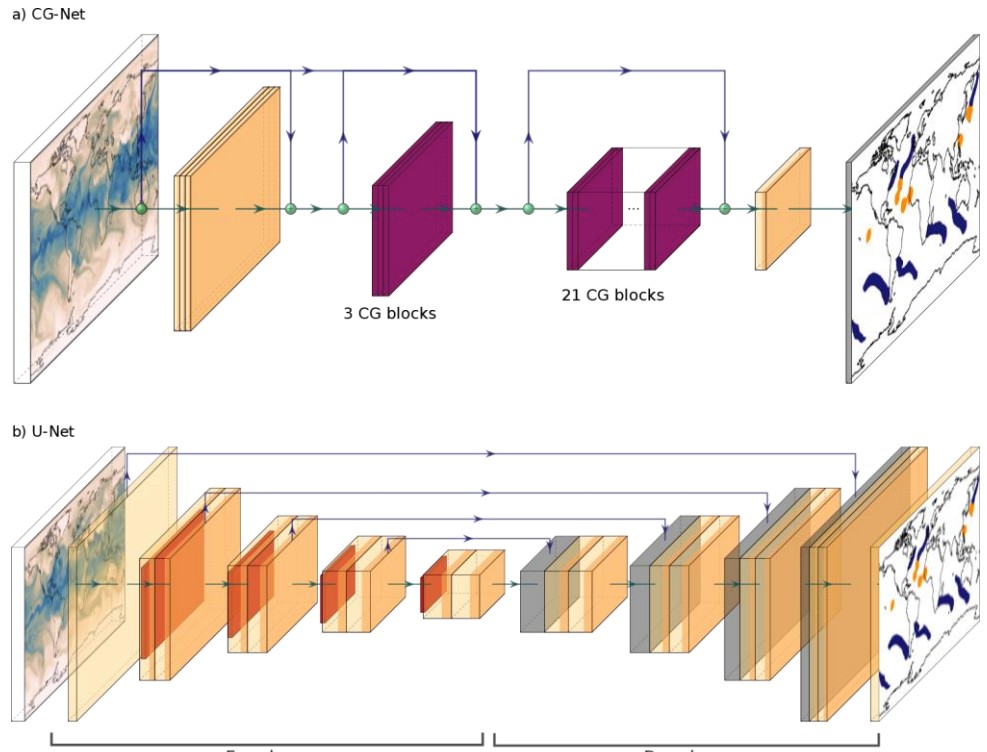

**Figure 1: Schematic illustration of the (a) CG-Net (Wu et al., 2021) and (b) U-Net (Ronneberger et al., 2015) ANN architectures. Yellow colour denotes convolutional layers, red average pooling layers, blue/grey transposed convolutional layers, violet context guided blocks. Blue arrows indicate skip connections.**

We use the same CG-Net configuration used by KS20, who in turn followed Wu et al. (2021). Our U-Net configuration is based on Ronneberger et al. (2015). For training CG-Net and U-Net, we follow KS20. Most grid points in the ClimateNet dataset belong to the background class, hence an imbalance exists between the frequency of the three classes. KS20 use the Jaccard loss function beneficial in cases of class imbalance (Rahman and Wang, 2016). It applies the Intersection over Union (IoU; Everingham et al., 2010) metric commonly used in semantic segmentation (e.g., Cordts et al., 2016; Zhou et al., 2017;

Abu Alhaija et al., 2018). The IoU score characterizes the overlap of two features by dividing the size (in the computer



vision literature as number of pixels, in our case in grid points) of feature intersection by the size of feature union[3]. If two features are identical, the IoU score equals 1, if they do not overlap at all, the score equals 0. The Jaccard loss function is minimized (equivalent to maximising IoU) using the Adam optimizer (Kingma and Ba, 2014), with a learning rate of 0.001. Since random weight initialization leads to differing results in different training runs (Narkhede et al., 2022), we train each

network five times and select the best performing. We use convolution kernels of size 3x3 grid points. Grid boundaries in longitudinal direction are handled with circular (i.e., cyclic) padding; at the poles, replicate padding is used.

KS20 as well as P21 use only a subset of the 16 variables contained in the ClimateNet dataset: TMQ, U850, V850, and PSL in KS20, and TMQ, U850, V850, and PRECT in P21. We reproduce the KS20 CG-Net setup for our objective of investigating whether it bases its detection on plausible patterns. For evaluation, IoU scores are computed for each feature

class individually and for comparison with values provided by KS20 as multiclass means. All scores are listed in percent.

**Table 2: Intersection over union scores reached by CG-Net trained as proposed by KS20, using a batch size of 4 and 10. For comparison, scores for U-Net are provided. All values are in percentages. The highest score per column is written in bold.**

| CNN implementation | AR | TC | AR-TC Mean | BG | AR-TC-BG Mean |
|---|---|---|---|---|---|
| CG-Net implementation by KS20, batch size 4 | **40.8** | 35.3 | 38.0 | 94.1 | 56.7 |
| CG-Net implementation by KS20, batch size 10 | 40.3 | 35.9 | **38.1** | 94.4 | 56.8 |
| U-Net (Ronneberger et al., 2015), batch size 4 | 40.2 | 36.0 | **38.1** | 94.3 | 56.8 |
| U-Net with num. of neurons per layer reduced to ¼ of Ronneberger et al. (2015), batch size 10 | 40.1 | **36.1** | **38.1** | **94.7** | **57.0** |

Table 2 lists evaluation results for both CG-Net and U-Net. KS20 only provide an evaluation score for the AR-TC-BG mean of 56.1%. Our reproduction (using the same implementation) yields a similar score (slightly different due to random initialization); Table 2 in addition shows the scores for the individual feature classes. KS20 use a training batch size of 4; with 20 training-evaluation epochs a single training run takes about 19 min on an 18-core Intel Xeon® Gold 6238R CPU with 128 GB RAM and a Nvidia A6000 GPU with 48 GB VRAM. To speed up training, the batch size can be increased. For

instance, a batch size of 10 reduces training time for a single run by 10% while achieving similar evaluation results. The U-Net implementation achieves similar scores, confirming that the detection task can be learned by different CNN architectures. Also, our experiments showed that for U-Net, reducing the number of neurons per layer to one quarter compared to the original Ronneberger et al. (2015) implementation reduces training time by 35% while achieving similar

---

[3] Note that this approach does not entirely correspond to the geometric area of the features on the globe. For features that occur closer to the poles a metric based on the geometric feature area would be more suitable.



evaluation results. One may hypothesize that due to its larger number of weights the U-Net architecture has an increased

potential to learn complex tasks and thus may achieve higher IoU scores for the problem at hand. This, however, seems not

to be the case. Also, U-Net in our case requires 50 training-evaluation epochs to converge, requiring about 46 min on our

system.

Concluding, all CNN setups achieve very similar evaluation scores, which provides confidence that they are learning similar

structures that can be further analysed using LRP.

**4.     Adapting LRP to semantic segmentation**

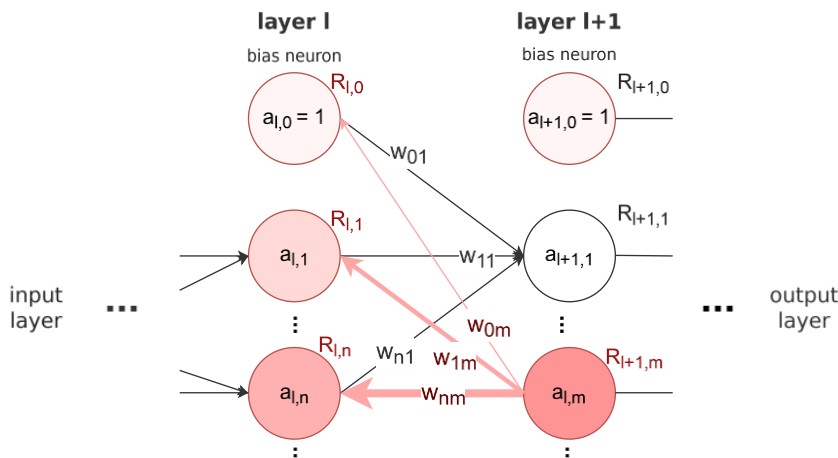

**Figure 2: Schematic illustration of two (hidden) layers of an ANN (cf. Eq. 1). $a_{l,m}$ denotes the activation of neuron $m$ in layer $l$, $R_{l,m}$ the corresponding relevance, $w_{n,m}$ the weight between neuron $n$ and $m$. Neuron "0" of each layer is a bias neuron. Red colour intensity symbolises exemplary relevance backpropagated from neuron $m$ in layer $l+1$ towards layer $l$, distributed according to**

**neuron activation and weights. In setups discussed in this study, activation $a$ and relevance $R$ are 3-D grids with size of the current layer-dependent horizontal grid times the number of classes; the weights $w$ can be scalars or convolution kernels depending on layer type.**

For our first objective, we adapt LRP to the semantic segmentation task. LRP was originally developed to understand the

decision-making process of ANNs designed for solving classification tasks (Bach et al., 2015). After a classification-ANN

has computed class probabilities from some input data grid, LRP considers a single feature class by only retaining its

probability (all other class probabilities are set to zero). This modified output is interpreted as initial value for the relevance

to be computed; it is propagated backwards through the network towards the input layer. Figure 2 illustrates the approach. In

an iterative process, the relevance $R_{l+1,m}$ of a given neuron $m$ in a network layer $l+1$ is distributed over all neurons $n$ in the



preceding layer $l$ (conserving the total relevance). Practically, this is implemented by iteratively computing the relevance $R_{l,n}$
of the neurons in layer $l$, as proposed by Montavon et al. (2019):

$$\mathbf{R_{l,n}} \; = \; \sum_{\mathbf{m}} \frac{\mathbf{a_{l,n} \cdot \rho\left(w_{l,n}^{(l+1),m}\right)}}{\boldsymbol{\epsilon} + \sum_{\mathbf{n}} \mathbf{a_{l,n} \cdot \rho\left(w_{l,n}^{(l+1),m}\right)}} \mathbf{R_{l+1,m}} \tag{1}$$

Here, $a_{l,n}$ denotes the activation value of neuron $n$, $w_{l,n}^{(l+1),m}$ the weights between neurons $n$ and $m$, $\rho$ an optional function
that modulates the weights, and $\epsilon$ a constant value that can be used to absorb weak or contradictory relevance. In our case,
activation $a$ and relevance $R$ are 3-D grids with size of the current layer-dependent horizontal grid times the number of
classes; the weights $w$ can be scalars or convolution kernels depending on layer type. We refer to Montavon et al. (2019) for
further details. In this study, we use the so-called LRP$_z$ rule (M22; also called LRP-0 rule; e.g., Montavon et al., 2019), i.e.,
the function $\rho$ is a simple identity mapping, and $\epsilon = 1 \cdot 10^{-9}$ to prevent division by zero. The relevance distribution of the input
layer is the desired result. LRP$_z$ distinguishes between positive and negative contributions, which can be interpreted as
arguments *for* (positive relevance) and *against* (negative relevance) classifying grid points as belonging to a feature.
Other LRP rules exist, M22 discussed their properties for CNN architectures designed for classification (note that the
LRP$_{comp}$ and LRP$_{comp/flat}$ rules recommended by M22 are not directly applicable to our setup; e.g., our CNNs do not contain
fully connected layers; also, the LRP$_{comp/flat}$ rule cannot distinguish between different input variables).

As noted in Sect. 2, LRP using the LRP$_z$ rule suffers from what M22 called the "ignorant-to-zero-input issue". The ANN's
bias neurons (e.g., Bishop, 2007; index 0 in each layer in Figure 2), required, e.g., to consider input values close to zero that
would otherwise have no effect on the ANN output due to the multiplicative operations at each neuron (e.g., Saitoh, 2021).
Due to the design of LRP$_z$, relevance assigned to bias neurons is not passed on to the previous layer and will not be included
in the final result. Hence, input values of zero will receive zero relevance (Montavon et al., 2019).

LRP implementations for classification tasks have been described in the literature (e.g., Montavon et al., 2019; M22). For
semantic segmentation tasks, the question arises how the gridded output (instead of single class probabilities) should be
considered. A straightforward approach is to consider an individual detected feature (i.e., a region of connected grid points of
the same class) and to compute a relevance map for each grid point of the feature, i.e., treating each grid point as an
individual classification task. Then, the resulting relevance maps can be summed to obtain a total feature relevance.

An important aspect is that the absolute relevance values computed by LRP depend on the absolute probability values
computed by the CNN. For example, if a grid point is classified as TC based on probabilities (TC=0.3, AR=0.2, BG=0.1),
the corresponding relevance map will contain lower absolute relevance values than if the probabilities were, e.g., (TC=0.8,
AR=0.6, BG=0.4). The question arises whether the relevance values should be normalized before summation, as the absolute
probability values are not relevant for assigning a grid point to a particular class. They can, however, be interpreted as how
"certain" the CNN is for the assignment. We hence argue that for all grid points belonging to a given feature, no
normalization should be applied. This way, in the resulting total relevance map the individual grid points' contributions are



weighted according to their probability of belonging to the feature, higher relevance is deemed to be more important for the overall feature as well.

To compare relevance maps of distinct features, or to jointly display the relevance of multiple features in a single map, we however argue that the relevance maps of the individual features should be normalized first. This ensures that the spatial structures relevant for the detection of a feature show up at similar relevance magnitudes.

For computing relevance maps for the individual grid points an existing LRP implementation for classification, e.g. Captum (Kokhlikyan et al., 2020), can be used by adding an additional layer to the network that reduces the output grid to a single point (this corresponds to setting the output probabilities of all grid points except the considered one to zero). This approach has recently been used by Farokhmanesh et al. (2023) for an image-to-image task similar to semantic segmentation. The resulting relevance maps can in a subsequent step be summed and normalized. Depending on grid size and number of

neurons in the CNN, this approach, however, can be time-consuming (in our setup, a single LRP pass requires about 100 ms; with an AR feature typically consisting of more than 5000 grid points in the given dataset, calculating LRP with this approach sums to about 8 min for an AR feature). To speed up the computation, we modify the approach by retaining the output probabilities of all grid points that belong to a specific feature. The LRP algorithm is executed only once (thus only requiring about 100 ms for an entire AR feature), due to the distributive law for addition and multiplication, this is equivalent

to the first approach. This approach has also been used by Ahmed and Ali (2021) for a specific U-Net architecture in a medical application, although for the entire data domain instead of individual features.

To apply LRP with the CG-Net architecture, the additional challenge of handling CG-Net-specific layer types arises. In addition to layer types also present in the U-Net architecture (for which LRP implementations have been described in the literature, including convolutional layers, Montavon et al., 2019; pooling layers, Montavon et al., 2019; batch normalization

layers, Hui and Binder, 2019; Guillemot et al., 2020; and concatenation-based skip connections, Ahmed and Ali, 2021), CG-Net uses addition-based skip connections, a spatial upscaling layer, and a global context extractor (GCE; Wu et al., 2021).

For addition-based skip connections, we first calculate the relative activations of both the skip connection and the direct connection in relation to the summed activation. Next, the relevance of the subsequent deeper layer is multiplied with these relative activations to determine the relevance for both connections. LRP for spatial upscaling layers is calculated by

spatially downscaling the relevance maps by the corresponding scaling factor. Following the argumentation by Arras et al. (2017) for adapting LRP to multiplicative gates in Long Short-Term Memory (LSTM) units, we omit the relevance calculation of GCE units.



# 5. Case study: Plausibility of spatial relevance patterns for detected TC and AR features








**Figure 3: Global maps of (a) TMQ and (b) PSL for the time step contained for 27 September 2013 in the ClimateNet dataset. Orange (red) contours show TC (AR) features detected with CG-Net using the KS20 setup. Panels (c) and (d) show z-score normalized fields input to the ANN. Panels (e) and (f) show summed TMQ and PSL relevance of all grid points classified as TC, panels (g) and (f) the summed relevance of all grid points classified as AR.**

For our second objective, we discuss the example of 27 September 2013 and assess the plausibility of spatial relevance patterns obtained using our adapted LRP approach and the CG-Net setup that reproduces the KS20 setup (using z-score normalization; cf. Table 2). On the chosen day, several TC and AR features were present that we consider representative.

Figure 3a/b shows global maps of TMQ and PSL of the chosen time step, overlaid with detected TC and AR features. Distinct features in the North Atlantic region are enlarged. In general, TCs are characterized by high humidity and minima in

PSL (e.g., Stull, 2017), ARs by strong horizontal moisture transport (implying high humidity and wind speed; e.g., Ahrens et al., 2012). ARs also take the form of elongated bands of elevated humidity connected to mid-latitude cyclones (Gimeno et al., 2014). These aspects are commonly used by rule-based detection methods (e.g., Tory et al., 2013; Shields et al., 2018; Nellikkattil et al., 2023), we are hence interested in whether CG-Net learns similar aspects.

In addition to the colour-coded TMQ and PSL fields, Fig. 3c/d also shows the z-score-normalized fields that are the actual

input to the ANN. As discussed in Sect. 4, the employed $LRP_z$ rule is "ignorant to zero input" (M22), it is hence important to see where zero values are input to the CNN.

Figure 3e/f shows the relevance of TMQ and PSL for the detected TC features (i.e., the summed relevance of all grid points classified as TC as described in Sect. 4). We interpret the relevance maps as "what the CNN looks at" to detect a feature, and where it collects arguments *for* (positive relevance) and *against* (negative relevance) classifying grid points as belonging to a

feature. If the relevance is close to zero *despite having a non-zero input value*, the corresponding location is considered irrelevant to the current feature of interest.

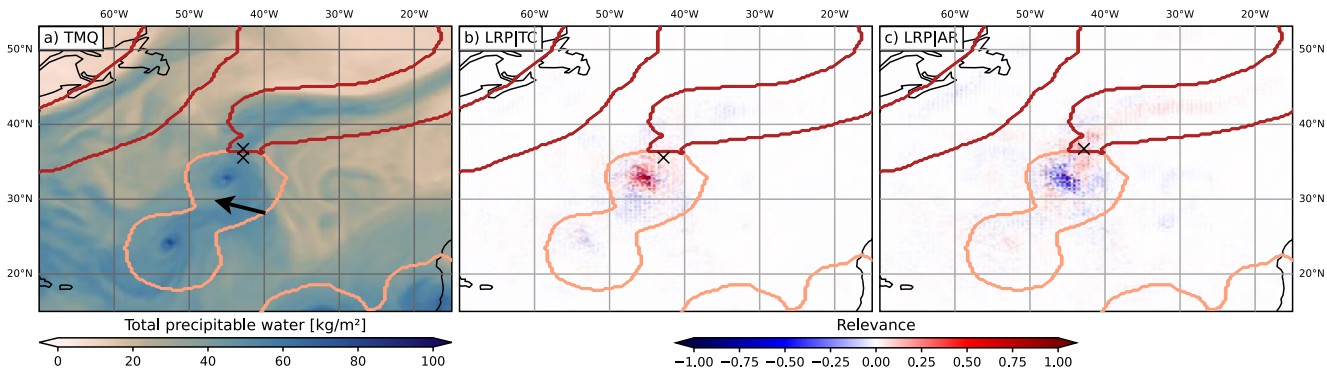

**Figure 4: (a) Close-up of TMQ as shown in Fig. 3a for the North Atlantic region. Black "x" mark two selected single grid points,**

**the southern one within the TC feature, the northern one within the AR. (b) TMQ relevance for the southern (TC) grid point. (c) TMQ relevance for the northern (AR) grid point.**





For both TMQ and PSL, CG-Net learns to positively consider extreme values at the centre of the detected TCs, with TMQ considered more relevant than PSL (normalized relevance of up to 1.0 vs. up to 0.5). The relevance mostly is spatially confined to the feature region. Positive TMQ relevance is mostly found at TMQ maxima, which also correspond to z-score

normalized maxima (Fig. 3a/c). PSL relevance is also collocated with PSL minima, which, however, are surrounded by bands of close-to-zero values after z-score normalization. We hypothesize that this can cause the lower relevance values compared to TMQ, i.e., CG-Net could consider PSL values more strongly, but this is not discernible in the $LRP_z$-computed relevance of the used setup.

A further noticeable characteristic in Fig. 3e/f is that the detected TC features are markedly larger than the relevant regions.

Here our hypothesis is that CG-Net learnt to classify grid points at a certain distance around point-like extrema as TC, i.e., for grid points at the edge of a feature the most relevant information is that it is at a specific distance to the TMQ maximum and PSL minimum. This hypothesis would be consistent with the specific capabilities of CNNs; their convolution filters take neighbouring grid points into account (e.g., Bishop, 2007). If CG-Net had primarily learned some sort of *thresholding* on TMQ or PSL, and no information about the *spatial structure* of the fields, we would have expected the relevance to cover the

feature area (with values above/below a specific threshold) more uniformly.

To test the hypothesis, we consider two individual grid points in the inset region in Fig. 3 that are of interest because they are close to the border of the TC and AR features around 45° W and 35° N: How does CG-Net distinguish between the two feature classes in this region? Figure 4 shows TMQ relevance maps for the two points, the southern one being classified as belonging to the TC, the northern one as belonging to the AR (black crosses in Fig. 4a, note that in Fig. 4b/c the relevance

for the classification of the single grid points only is shown, not the summed relevance of all feature grid points as in Fig. 3). For the TC grid point, Fig. 4b shows that the ANN considers the nearby TMQ maximum at 43° W and 31° N as a strong argument *for* its decision to classify the point as TC, confirming our hypothesis. Some patches, in particular south of the TC centre, are considered as arguments *against*, though at much weaker relevance magnitude. We hypothesize that this may be due to the *shape* of the TMQ field in this region with weak filaments of TMQ being drawn into the TC from south-west

(arrow in Fig. 4a); we will come back to this issue in the next section. For the AR grid point, CG-Net considers the nearby TMQ maximum as a strong argument *against* classifying the point as AR. In contrast, the also nearby band of high TMQ extending from 40° W and 40° N towards north-east is considered as an argument *for* the point being part of an AR. We interpret these findings such that the CNN indeed considers the spatial distance to a point-like TMQ maximum, and possibly also the filamentary structures in TMQ. Note that the final classification decision, however, is of course based on all input

fields.



**Figure 5: Same as Fig. 3 but for (left column) zonal wind at 850 hPa and (right column) meridional wind at 850 hPa.**






Figure 3g/h shows the relevance of TMQ and PSL for the detected AR features. We again focus on the North Atlantic region in the inset, containing two ARs. The elongated band of high TMQ associated with the eastern AR is surrounded by dryer air, making it distinctly stand out in Fig. 3a. CG-Net finds positive relevance in this band, few arguments against the structure being an AR are found in its surroundings except for the discussed TMQ maximum in the TC directly south of the AR (Fig. 3g). However, again we note that if information directly around the band of high TMQ was considered by the CNN, it would not show up in the relevance map as the z-score-normalized values surrounding the band are close to zero (Fig. 3c). The western AR, however, is not as clearly surrounded by drier air and hence not as clearly discernible in the TMQ field (Fig. 3a). While for this feature also the elongated band of high humidity is taken as an argument *for* the AR class, south of the AR regions of arguments *against* show up (Fig. 3g). We interpret this as some sort of uncertainty of the CNN, like a human expert that would analyse the region around this AR more carefully, also considering other available variables to make their decision.

Figure 3h shows that for AR detection CG-Net cannot infer much information from the PSL field. While it "looked" at the regions surrounding the ARs, the relevance field is weak and noisy, and no recognizable structure is found. This is plausible since in Fig. 3b there are no discernible PSL structures visible for the AR features. To the best of our knowledge, there are also no rule-based systems that use PSL for AR detection.

Figure 5 shows, however, that both 850 hPa wind components are used for the detection of both TCs and ARs. The western AR in the inset is characterized by high zonal wind (Fig. 5a), which coincides with a clearly positively relevant structure (Fig. 5g). Meridional winds are strongest around the mid-latitude cyclone at the northern end of the AR (Fig. 5b), CG-Net also considers this as positively relevant (Fig. 5h). The dipole structure discernible in both wind components close to cyclone centres (both tropical and mid-latitude) is considered as an argument against ARs by the CNN (negative relevance in Fig. 5g/h). Our interpretation is that CG-Net learnt to identify such dipoles with TCs and cannot infer that a mid-latitude dipole north-east of an AR would be an argument *for* the AR feature. This is supported by that for detection of TC features, the dipoles are considered positively relevant (Fig. 5e/f). Both wind components are also widely used in rule-based detection systems, for example by three of the four algorithms discussed by Bourdin et al. (2022) for detection of TCs. For rule-based AR detection, wind components are contained in the integrated vapor transport (IVT) variable that is commonly used (Wick et al., 2013; Shields et al., 2018).

We conclude the discussion with the interpretation that CG-Net in the present setup learnt overall very plausible structures to detect TCs and ARs, which is very promising for gaining confidence in CNN-based detection of atmospheric features.

Reproductions of Figs. 3 and 5 when using U-Net instead of CG-Net are provided in the Supplement (Figs. S6 and S7). Despite the differences in CNN architecture (cf. Sect. 3), very similar results are found. Notable differences include that the U-Net setup detects smoother feature contours, and that its relevance values are more pronounced and show smoother spatial patterns. We consider it promising, however, that two different CNN architectures learn very similar patterns.



## 6. Relevant input variable values: The issues of shape and input normalization for explaining feature detection

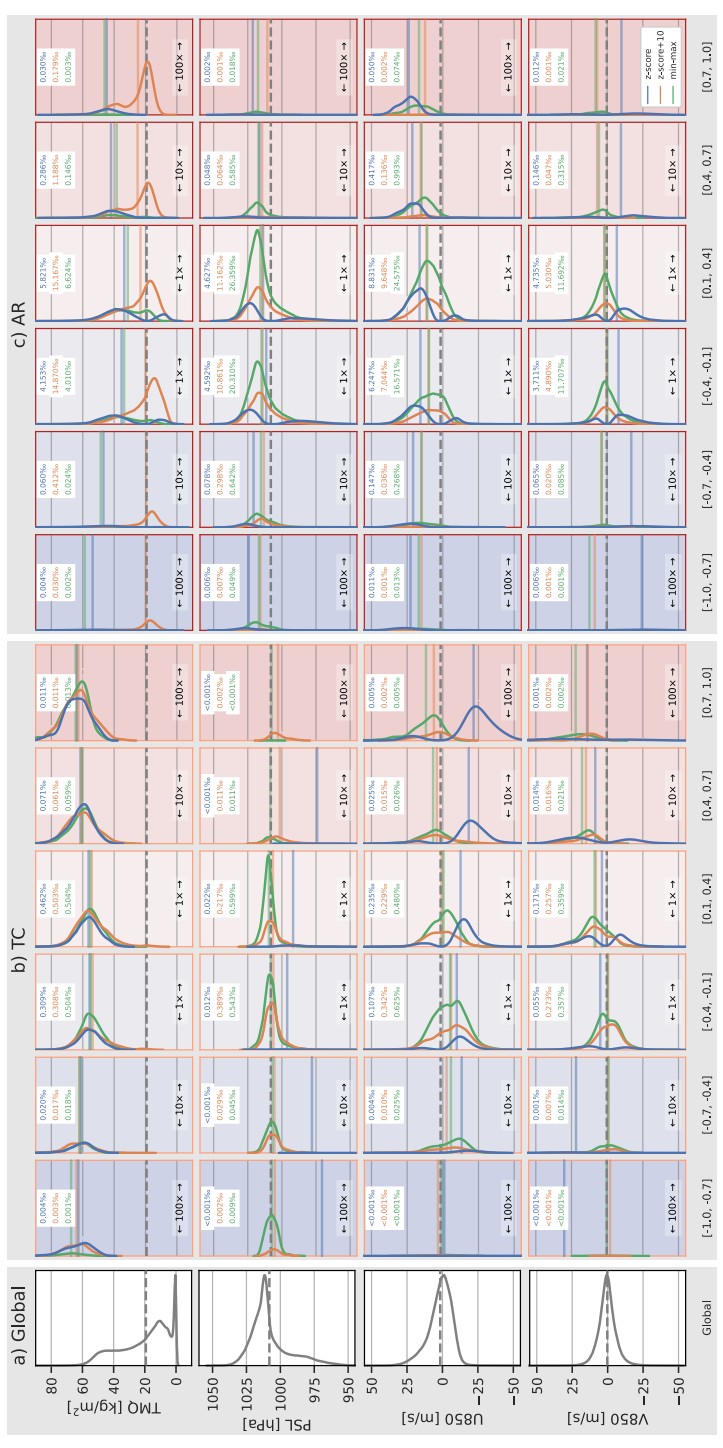

 

**Figure 6: Distributions of CG-Net input variable values in the test dataset. (a) Global distribution (all grid points). Dashed**
**horizontal lines show distribution means, for reference also shown in the other panels. (b) Distributions of grid points with**
**relevance magnitude > 0.1 for TC detection, at six different relevance ranges. The range [-0.1..0.1] is omitted. Shown are**
**distributions for z-score-normalized input values (blue curves), shifted z-score-normalization (orange), and min-max-**
**normalization (green). Horizontal lines show distribution means. The numbers at the top of each box denote fraction (in ‰) of grid**
**points in the corresponding relevance range. Note the horizontal scaling: Since much fewer grid points are assigned high relevance**
**values, to see the shape of the distributions we horizontally scale the relevance range [0.4..0.7] 10 times and the range [0.7..1.0] 100**
**times compared to the range [0.1..0.4]. (c) Same as (b) but for AR detection.**

Figures 3 to 5 showed a single time step that we consider representative as an example of the spatial relevance patterns
obtained from LRP. For a more complete picture of what the CG-Net has learnt, we are interested in statistical summaries of
input values that it considers relevant. The goal is to see if, for example, also on average high values of TMQ are learnt to be
most relevant for TC and AR detection.

We compute distributions over all time steps in the test dataset (cf. Sect. 2) of the CNN input variables, both at all grid points
and at grid points considered relevant to different extent (note that, as seen in Figs. 3 and 5, this includes grid points outside
the detected features). Figure 6 shows the distributions of TMQ, PSL, U850, and V850. As reference, the value distributions
for the entire globe, i.e., all grid points, are shown (Fig. 6a). To learn which variable values are considered relevant by LRP
for TC and AR detection, we divide the relevance range [-1..1] into six distinct intervals of width 0.3 and show distributions
for each feature and interval. The relevance range [-0.1..0.1] is omitted to mask out regions of zero and low relevance. Note
that for all variables, this includes over 98% of all grid points, i.e., on average less than 1% of all grid points are assigned
relevance values with magnitude larger than 0.1 in the present CG-Net setup. Distributions of the relevance range [-0.1..0.1]
hence look very similar to the reference distributions of the entire globe.

First notice that CG-Net for TC detection, also averaged over the entire test dataset, considers high values of TMQ positively
relevant. Already for the relevance range [0.1..0.4], the distribution peaks at values slightly below 60 kg m$^{-2}$, which is at the
upper end of the global distribution. The TMQ distributions of grid points with higher relevance peak at even slightly higher
values (although much fewer grid points are assigned high relevance). This finding is in line with our hypothesis from Sect.
5 that CG-Net learnt to associate TCs with TMQ extrema.

It is noticeable that the distributions of positive and negative relevance intervals cover similar TMQ values, however, with
more relevant grid points on the positive side. This raises the question why similar values are considered in both pro and
contra arguments – GG-Net could have also learnt to use *low* TMQ values as an argument against a TC feature. However,
high TMQ values not only occur within TCs but also elsewhere particularly in the tropics (cf. Fig. 3a). In Sect. 5 we discuss
that CG-Net is capable of learning spatial structure by means of convolution filters. We hence hypothesize that the
"pro/contra TC decision" is based on spatial structure, which cannot be inferred from the relevance distributions in Fig. 6.



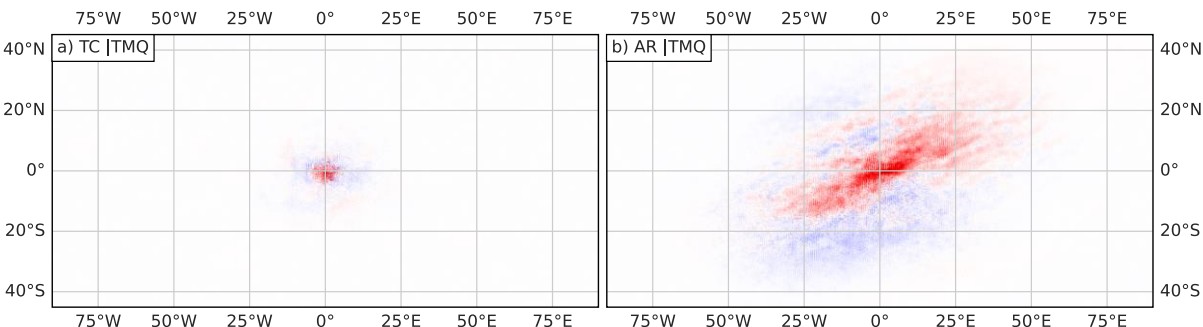

**Figure 7: Composite relevance maps for TMQ, averaged over (a) all TC features in the test dataset, and centred on the features. (b) The same for ARs. For clarity of presentation only AR features from the northern hemisphere are composited due to the difference in orientation on the southern hemisphere.**

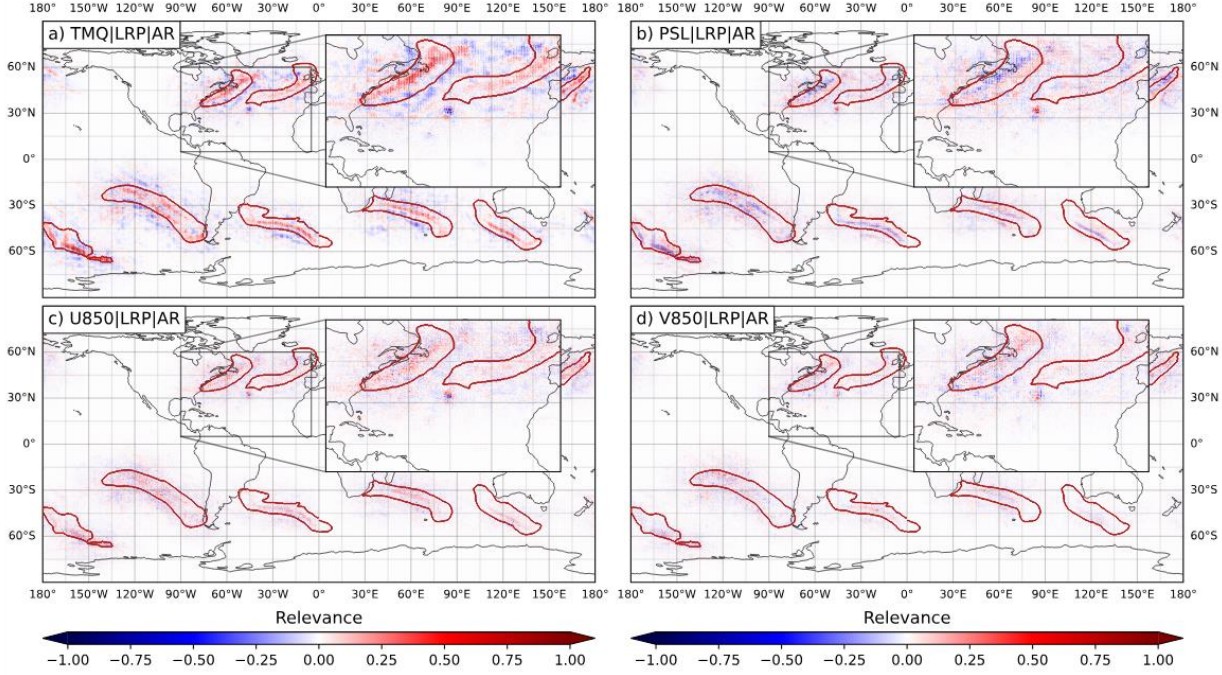

**Figure 8: Same as Fig. 3g/h and Fig. 5g/h, but for relevance obtained from CG-Net trained with z-score-normalized input data shifted by +10.**

First consider the distributions shown with the blue curves in Fig. 6b. They correspond to the KS20 setup with z-score normalized data used in the previous sections. Most distributions of relevant grid points clearly differ from the global reference distributions; hence CG-Net gathers information from the values of the input variables.





To investigate, we compute composite relevance maps of all TC and AR features in the test dataset. Figure 7 shows the
average relevance of TMQ for both feature classes, obtained by averaging the relevance of all features. For clarity of
presentation, only AR features from the northern hemisphere are considered since their orientation differs between both
hemispheres (if ARs from both hemispheres are plotted we obtain a cross-shaped pattern). Figure 7 shows that for TCs, CG-
Net on average learnt to detect spherical structures, while for AR elongated structures from south-west to north-east are
detected (north-west to south-east on the southern hemisphere; not shown). We interpret this finding as strong support for the
hypothesis that spatial structure plays a crucial role in the detection process, however, note that more detailed investigation is
required. Shape information is not directly accessible via LRP, and to the best of our knowledge not much literature has
investigated the explanation of shapes in CNN-based feature detection in general. While recently a potentially useful method
(Concept Relevance Propagation, CRP; Achtibat et al., 2023) has been published, it has yet to be applied to meteorological
data and is left for future work.

Figure 6c shows the TMQ distributions of grid points relevant for AR detection. While the distributions also show that more
relevant grid points correspond to higher TMQ values (which is plausible given the discussion in Sect. 5), we here observe
bi-modal distributions with minima located around the mean of the global distribution. As discussed in Sects. 4 and 5, values
around the global mean become close to zero after z-score-normalization, hence the minimum could be a consequence of the
"ignorant-to-zero-input-issue" (M22). The question hence arises whether CG-Net indeed does not consider TMQ values
around the global mean of 19.3 kg m$^{-2}$ (cf. Table 1; for which to the best of our knowledge there would be no plausible
physical reason), or whether that relevance information is simply missing in LRP$_z$ output.

To investigate, we re-train CG-Net with two alternative normalizations. First, we shift the z-score-normalized data by +10.
The value of 10 is chosen as minimum values after z-score normalization are about -8, hence the shift ensures that all input
data are positive and at some distance from zero. Second, we apply the min-max normalization (e.g., García et al., 2014) that
linearly scales all inputs to the positive range [0..1]. It, however, has the disadvantage of being more sensitive to outliers and
cannot ensure that all inputs are treated equally important by the CNN (since the means of the different input variables are
not mapped to the same normalized value; cf. Sect. 2).

IoU evaluation scores for both alternative normalizations are comparable to the original z-score normalization (e.g., AR-TC
mean of 37.8 for z-score+10 and 37.7 for min-max, compared to 38.1 for the original z-score setup). Also, Fig. 6c clearly
shows that with the alternative normalizations, TMQ values around the global mean *are* attributed to be relevant; the
minimum in the z-score-normalization distribution vanishes and in particular for the z-score+10 data, a maximum is found
instead. Hence, CG-Net *does* consider TMQ values in this range relevant.

Figure 8 revisits the case from Sect. 5 and shows AR relevance maps obtained from the CG-Net trained with z-score+10-
normalized inputs. Full reproductions of Figs. 3 and 5 for both alternative normalizations are provided in the Supplement
(Figs. S2-S5). Figure 8a shows that, compared to Fig. 3g, the elongated AR bands of high TMQ are still distinctly positively
relevant, but some noisy relevance is now found in the surroundings of the ARs, exactly where TMQ values around the





global mean are found. We interpret this finding as further confirmation that CG-Net does consider the *values* of TMQ for feature detection, but only in combination with *shape* information.

For min-max normalization, similar results are found for the case from Sect. 5 (cf. Supplement, Figs. S4-S5), although the relevance of TMQ values around the global mean is not as pronounced as for the shifted z-score normalization (Fig. 6c). For TC detection, the TMQ distributions hardly differ for the three normalizations (Fig. 6b), in this case, however, the relevant TMQ values are all well above the global mean (and hence already for the original z-score-normalized data above zero).

We find similarly plausible results for the other input variables. Notably, with the alternative normalizations the PSL input also shows up as relevant for TC detection (Fig. 6b); for AR detection, the PSL distributions become unimodal as for TMQ. With the alternative normalizations, however, the distributions of relevant PSL values are very similar to the global distribution, indicating that CG-Net does not infer much information from PSL *values*. Since, however, the number of relevant grid points is of the same order as for the other input variables (cf. the fractions listed in Fig. 6b/c), CG-Net *does* use PSL inputs – likely using *shape* information from this field. Further evidence for this hypothesis is found in Fig. 8b, where PSL relevance also shows elongated structures aligned with the ARs.

For the U850 and V850 wind components, the bi-modal distributions of relevant wind values obtained from the original CG-Net setup (both for TCs and ARs) could have been plausible in that the CNN only considers stronger winds. However, relevance from the alternative normalizations shows that also grid points with weak winds are considered relevant. An example of this is that in Fig. 8c/d the entire AR structures show relevance, including the regions of weak wind at the southern parts of both ARs (cf. Fig. 5a/b). This relevance is not present in Fig. 5g/h; the finding again suggests that shape information used by CG-Net. The distributions in Fig. 6c show, however, that for ARs, more relevant grid points are associated with elevated eastward winds (positive U850 component). This is plausible since on both hemispheres ARs are characterized by mid-latitude eastward winds.

Concluding, the obtained distributions also provide evidence that CG-Net learnt physically plausible structures for TC and AR detection. However, due to its inability to attribute relevance from bias neurons, LRP$_z$ applied to the original KS20 CG-Net setup using z-score normalization does not yield information about input values close to zero after normalization, which limits its use. Also, the use of shape information by the CNN cannot be attributed. Both information, however, would be required for full analysis of the learnt detection rules.

Again, reproductions of Figs. 6 and 8 when using U-Net instead of CG-Net are provided in the Supplement. For the U-Net setup, we observe that a larger number of grid points is considered relevant, however, the shape of the distributions (Fig. S8) remains similar to the CG-Net setup (Fig. 6). Also, changes in spatial relevance patterns when using the shifted z-score normalization instead of z-score normalization (Fig. S9) are analogous to the CG-Net setup (Fig. 8).





## 7. LRP applications: Finding most relevant input variables and evaluating detection robustness

In addition to providing a means to open the "black box" of a given CNN-based semantic segmentation setup as demonstrated in Sects. 5 and 6, we investigate further applications of LRP for semantic segmentation. Here, we discuss two

applications: (A1) Finding the most relevant input variables for a given feature detection task, and (A2) evaluating the robustness of a trained detection-CNN when some characteristic of the input data is changed, e.g., grid resolution is changed, or a different geographical domain is used.

Regarding A1, P21 provided 2-D fields of 16 atmospheric variables in the ClimateNet dataset (cf. Sect. 2). Today's numerical simulation models commonly output far more and also 3-D fields. KS20 and P21, on the other hand, used a subset

of four variables only to train their CNNs (TMQ, PSL, V850, U850 and TMQ, PRECT, V850, U850), M22 only used the three inputs TMQ, V850, U850. Using a subset of available variables can be beneficial, e.g., to reduce computational complexity (data acquisition and storage; computing time and memory requirements for CNN training) and to reduce overfitting issues when only limited training data is available (e.g., Schittenkopf et al., 1997). Suitable variables can be selected based on expert knowledge (using those variables that are known to be associated with the atmospheric feature of

interest, e.g., humidity and wind for TCs and ARs). However, how can suitable variables be selected if such knowledge is not readily available (e.g., for features not well investigated or if data for required variables are not available), without extensive evaluation of different variable combinations?

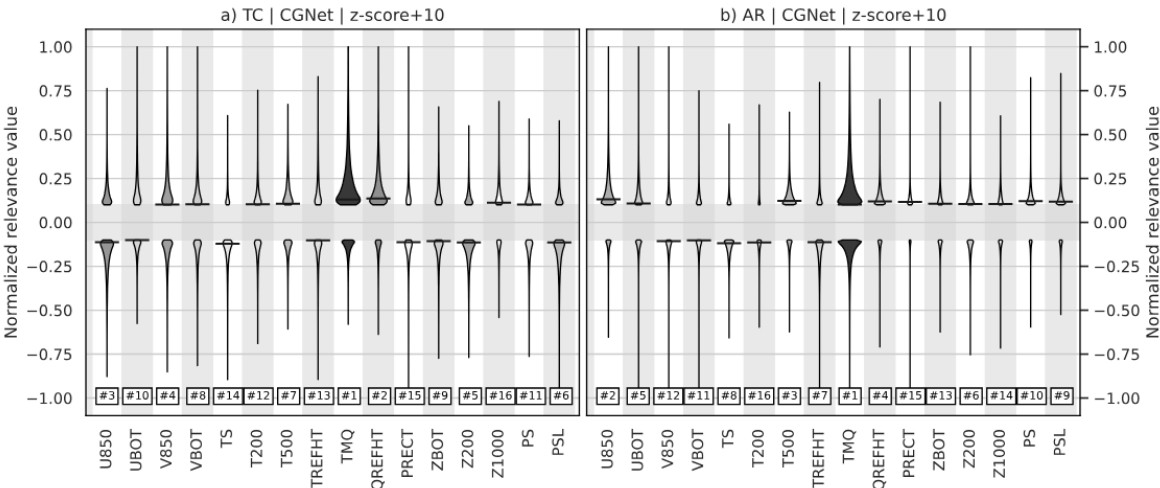

**Figure 9: Distributions of relevance values (computed from test dataset) for CG-Net trained with all 16 variables contained in the ClimateNet dataset, for (a) TC and (b) AR features. Z-score normalization shifted by +10 is used on all inputs for the reasons discussed in Sect. 6. Width of violin plots is differently scaled for TCs and ARs but consistent for all variables within (a) and (b). Relevance values in the range [-0.1..0.1] are omitted. Numbers at the bottom as well as grey shade indicate ranking in terms of numbers of grid points with absolute relevance >0.1.**





The analysis in Sect. 6 showed that for the different input variables, different fractions of grid points were found to be relevant in the different relevance intervals (numbers listed in Fig. 6). If a CNN is hence trained with *all* available input variables, distributions of relevance values can be computed for each input variable and the most relevant variables can be selected. We apply the approach to CG-Net trained with z-score-normalized inputs shifted by +10 (cf. Sect. 6), to avoid the "ignorance-to-zero-input-issue" (M22). Figure 9 shows violin plots (Hintze and Nelson, 1998) of the relevance distributions

for each of the 16 ClimateNet variables. As in Sect. 6, we omit absolute relevance below 0.1. Variables are shown in the same order as in Table 1, the given ranking is based on the number of grid points with absolute relevance larger than 0.1.

Indeed, TMQ is found to be the most relevant input variable for both TC and AR detection. For TCs, the QREFHT variable is also considered relevant by CG-Net, however, it should be closely correlated with TMQ. U850 and V850 are third and fourth, followed by several other variables of similar relevance. Some variables including TS, PS, and Z1000 are hardly of

relevance. For ARs, U850 is also considered relevant, however, V850 is not. The results suggest, however, that AR detection could benefit from including T500 in the set of input variables. These findings, of course, can be expected due to existing meteorological knowledge (Gimeno et al., 2014). It is promising, however, that LRP analysis again provides plausible results.

Table 3 shows IoU scores for CG-Net trained with all 16 input variables, the KS20 subset of TMQ, PSL, V850, U850, as

well as different selections that could be inferred from Fig. 9. The scores are largely of the same order, notably the three-input subset of TMQ, V850, U850 used by M22 achieves even higher scores than the KS20 subset and the 16-variable setup. Only when even more inputs are withdrawn the detection performance drops, although remains remarkably high. We note, however, that for every setup a relevance analysis as in Sects. 5 and 6 should be carried out to ensure plausible results.

**Table 3: IoU scores reached by CG-Net trained for different input variable combinations, using z-score normalization shifted by +10 for all input variables. All values are in percentages. The highest score per column is written in bold. Compare to Table 2.**

| Input variables | AR | TC | AR-TC Mean | Background | AR-TC-BG Mean |
|---|---|---|---|---|---|
| All 16 variables listed in Table 1 | **41.0** | 33.7 | 37.4 | **94.9** | 56.5 |
| TMQ-PSL-U850-V850 | 40.4 | 35.2 | 37.8 | 94.8 | 56.8 |
| TMQ-QREFHT-U850-V850-T500 | 40.3 | 35.5 | 37.9 | 94.7 | 56.8 |
| TMQ-T500-U850-V850 | 40.5 | 35.6 | 38.1 | 94.5 | 56.9 |
| TMQ-Z200-U850-V850 | 40.6 | **35.9** | **38.2** | 94.5 | **57.0** |
| TMQ-U850-V850 | **41.0** | 35.4 | **38.2** | 94.5 | **57.0** |
| TMQ-U850 | 40.4 | 33.8 | 37.1 | 94.6 | 56.3 |
| TMQ | 39.8 | 30.4 | 35.1 | 94.0 | 54.7 |



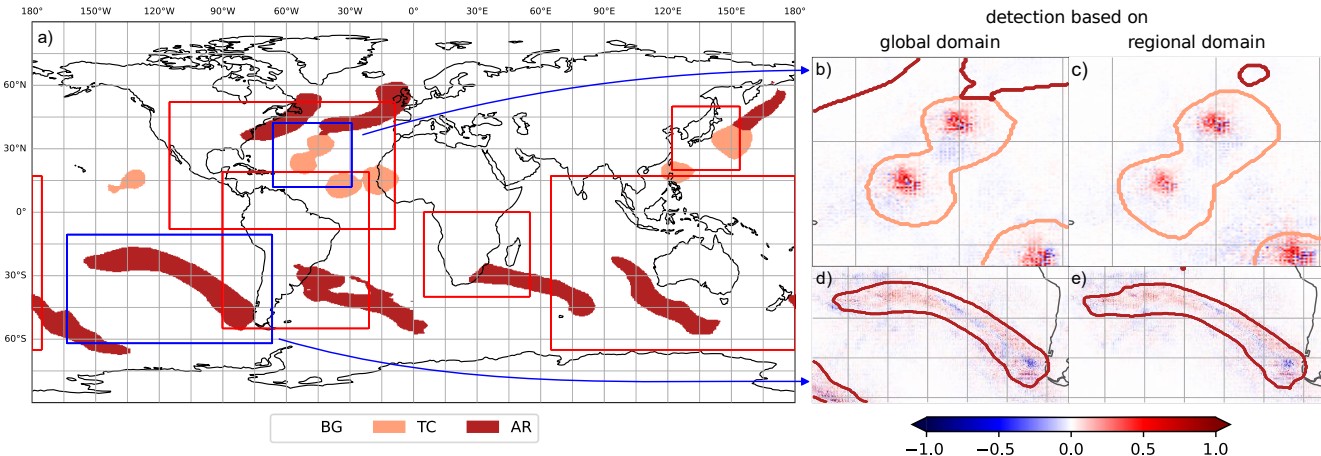

**Figure 10: Regional domains to evaluate the robustness of feature detection when the CG-Net trained on global data is applied with data on a regional domain. Same case and CG-Net setup (KS20 setup and z-score normalization) as in Figs. 3 and 5. Blue bounding boxes in (a) surround selected TC and AR features; spatial relevance patterns in these regions is shown for (b, d) global data input into CG-Net, then subregion cut out from global result, and (c, e) only subregion data input into CG-Net. Note that here relevance values are summed over all variables. Red bounding boxes in (a) show domains of regional NWP models: HAFS-SAR (North Atlantic), Eta (South America), SADC region (Southern Africa), MSM (Japan), ACCESS-R (Oceania). Domains are approximate where model domains are not rectangular in longitude and latitude.**

**Table 4: IoU scores for TCs and ARs detection in subregions used by different regional NWP models. CG-Net with the KS20 setup and z-score normalization is used (as for Figs. 3 and 5). Scores are computed for (a) global data input into CG-Net, then subregion cut out from global result, and (b) only subregion data input into CG-Net.**

| Region | (a) IoU of subregion (detection using global data) | | | (b) IoU of subregion (detection using regional data) | | |
|---|---|---|---|---|---|---|
| | AR | TC | BG | AR | TC | BG |
| Global (same as in Table 2) | 40.3 | 35.9 | 94.4 | | | |
| NOAA (North Atlantic) | 34.5 | 41.1 | 91.7 | 31.2 | 41.0 | 92.3 |
| CPTEC (South America) | 41.3 | 43.3 | 92.5 | 34.8 | 38.2 | 92.3 |
| SAWS (Southern Africa) | 39.9 | 5.0 | 91.6 | 25.5 | 0.0 | 90.9 |
| JMA (Japan) | 31.5 | 41.0 | 88.9 | 19.1 | 41.6 | 88.3 |
| BoM (Oceania) | 38.9 | 10.3 | 91.8 | 36.9 | 4.7 | 92.3 |

Regarding A2, consider that in both operational weather forecasting and atmospheric research, numerical weather prediction (NWP) models with a regional domain are frequently used. The analysis discussed in the previous sections was based on global data. Could the CG-Net trained on global data be also used with data from a regional domain, or would it have to be retrained? The analysis of spatial relevance patterns in Sect. 5 suggested that CG-Net mostly considers grid points within or



in close vicinity of a detected feature, hence we see a chance that detection with regional data could work "out of the box". This would be valuable for cases where CNN training is expensive (e.g., Niebler et al., 2022, reported high computational demand for training their front detection CNN), as a CNN trained globally could be applied to different regional models.

We consider our case from Sect. 5 and compare detected features and spatial relevance patterns (1) if the detection is based
on global data as in the previous sections, then the subregion is cut out, and (2) if the detection is based on regional data input into CG-Net trained on global data. For our experiments, we simply cut out data from the global ClimateNet data, i.e., grid point spacing is unchanged. Figure 10 shows how the detection result changes for two selected TC and AR features (blue boxes in Fig. 10a). Figure 10b/d shows the features and relevance patterns when global data is used, Fig. 10c/e when regional data is used for testing. Note that for simplicity of display, the relevance of all four input variables is averaged in
Fig. 10, and that for the regional domains, circular padding (cf. Sect. 3) is not suitable, here replicate padding is used instead. Figure 10b-e shows that the features at the centre of the regional domains are fully detected with high similarity between both approaches. In contrast, the features only partially included in the region are not or not completely detected. These findings are plausible given that due to the convolutional architecture of CG-Net, some area around a feature is required for detection. It is noticeable, however, that the inclusion of the TC centre in the south-eastern corner of Fig. 10b/c seems to be
sufficient to detect the partially included TC. This is also further evidence for our hypothesis from Sect. 5 that the distance to a TC centre plays a crucial role in the detection process. Also note that from the AR in the north-eastern part of Fig. 10b, a small part is still detected in the regional data (Fig. 10c). Unlike rule-based systems that often define a minimum size for an AR feature (Shields et al., 2018), CG-Net seems to not learn such size limitations.

The selected case provides promising evidence that indeed the CG-Net trained on global data could be used for detecting
features in regional data as well. For a more complete picture, we consider several regional domains used by national weather services (red boxes in Fig. 10a): National Oceanographic and Atmospheric Administration (NOAA) in the USA (Dong et al., 2020), Center for Weather Forecasting and Climate Studies (CPTEC) in Brazil (Alves et al., 2016), South African Weather Service (SAWS; Mulovhedzi et al., 2021), Japan Meteorological Agency (JMA; Saito et al., 2006), and Australian Government Bureau of Meteorology (BoM, Puri et al., 2013). Table 4 lists IoU scores for the respective regional
domains, again for feature detection based on (1) global data and (2) regional data. Despite using the same input data for detection, the IoU scores for (1) differ from the global IoU scores listed in Table 2 since only subsets of all features are present in the regional domains. Scores roughly deviate more from Table 2 for smaller subregions. For (2), IoU scores are lower compared to (1) for all subregions. This, however, is plausible considering the above discussion that features included only partially in a subregion are less well detected when only regional data is input to CG-Net. The differences in IoU scores
that we observe between approaches (1) and (2) are smaller for TCs (maximum difference of 5.6% in Oceania), and more substantial for ARs (difference of up to 14.4% for the South African region and 12.4% for the Japanese region). Our hypothesis is that this is due to the smaller size of TCs, which are hence more often completely contained in a subregion. Similarly, larger regional grids show higher IoU scores, possibly for the same reason of containing more complete features.



Concluding, we note that while detection performance decreases when regional data is used for testing, we argue that the

method still has value, e.g., to assist forecasters in becoming aware of potentially important features. Also, the issue of decreased detection performance for only partially contained features in a region also affects rule-based detection methods, e.g., if rules with respect to feature size are used.

Results for A1 and A2 when using U-Net instead of CG-Net are provided in the Supplement (Figs. S10 and S11; Tables S1 and S2). Both CNN architectures again yield very similar results. Notably, regarding A1, the U-Net setup also considers

TMQ to be the most relevant input, however, in contrast to CG-Net the TS input provides more information.

## 8.    Summary and conclusion

We adapted the xAI method Layer-wise Relevance Propagation, widely used in the literature for classification tasks, to be used for semantic segmentation tasks with gridded geoscientific data. We implemented the method for use with the CG-Net and U-Net CNN architectures (Fig. 1) and investigated relevance patterns these CNNs learnt for detection of 2-D tropical

cyclone and atmospheric river features. Our analysis built on previous work by KS20, P21, and M22. In this paper, we focused on the CG-Net setup suggested by KS20 using the four gridded and z-score-normalized input variables TMQ, PSL, U850, V850 from the ClimateNet dataset provided by P21 (Table 1). Comparative results for U-Net are provided in the Supplement.

The main findings from our study are:

• With both CG-Net and U-Net we were able to reproduce KS20/P21 results with similar IoU scores (Sect. 3; Table 2).

• Adapting LRP (Fig. 2) to the semantic segmentation task provided the challenge of how to generalize the classification approach used by previous studies. We argue that averaging relevance from all grid points assigned to a feature provides meaningful results. Also, to use our method with the CG-Net architecture, several layer-specific LRP calculation specifications had to be implemented for CNN layers specific to CG-Net (Sect. 4).

• For the selected case, we found that CG-Net learnt physically plausible patterns for the detection task (Sect. 5). For TCs, relevant patterns include point-shaped extrema in TMQ and circular wind motion; for ARs, relevant patterns include elongated bands of high TMQ with different orientation on northern and southern hemisphere, and eastward winds (Figs. 3 and 5).

• Spatial relevance is mostly locally confined around features, but analysis of the relevance of individual grid points

indicated that for each grid point, CG-Net uses its convolutional filters to account for the surrounding region (Fig. 4).

• CG-Net makes use of both input *values* and the *shape* of patterns in the input fields. Analysis of input variable values at grid points attributed high relevance showed that, e.g., high values of TMQ are relevant for both TC and AR detection, however, that these high values are used for both *pro* and *contra* arguments for assigning a grid point to a feature (Sect. 6; Fig. 6). This behaviour can be explained by the hypothesis that CG-Net uses additional shape information for its





decision. LRP does not provide information about shape relevance, however, composite maps we computed from all detected features provide strong evidence that TCs are detected as point-like structures and ARs as elongated bands (Fig. 7).

- Care needs to be taken when using LRP$_z$ with z-score normalization (mapping the mean of a variable to zero and its standard deviation to +/- 1; as used by KS20, P21 and M22). CNNs including CG-Net and U-Net include bias neurons to account for input data close to zero, however, LRP cannot attribute relevance to bias neurons. Hence, input values close to zero are assigned a relevance close to zero (referred to as the "ignorant-to-zero-input-issue" by M22; cf. Fig. 6), even if the CNN *does* use the information via the bias neurons. As a workaround, we shifted the z-score-normalized data by +10 to avoid zero values (and also evaluated use of min-max normalization that maps variable values to 0..1). With these alternative normalizations, zero relevance around variable means disappears (Fig. 6), and spatial relevance patterns further suggest the role of shape information in the detection process (Fig. 8).

- LRP can be used for additional applications (Sect. 7). We demonstrated its use for finding the most relevant input variables to build a CNN setup by training CG-Net with all 16 input variables in the ClimateNet dataset, then using relevance distributions to find the most relevant variables that need to be retained for a useful setup (Fig. 9 and Table 3). Also, we evaluated the robustness of detection when only data from subregions is used with the CG-Net trained on global data. This has potential benefit to use a globally trained CNN for detecting features in data from regional NWP models. We find that due to the locality of relevance, features fully included in a subregion are well detected, while only partially contained features are not (Fig. 10 and Table 4).

Concluding, LRP in our opinion is a very useful tool to gain confidence for CNN-based detection of atmospheric features. For the case of TC and AR detection proposed by KS20 and P21, we find that their setup indeed learns physically plausible patterns for feature detection. We provide the source code of our implementation along with this paper and invite the geoscientific science community for apply the method to further detection tasks. However, the open challenges of accounting for the relevance of bias neurons ("ignorant-to-zero-input-issue"; M22) as well as for shape information need to be approached to be able to explain the behaviour of CNNs for semantic segmentation tasks more completely. First work for accounting for bias relevance has recently been published in the computer vision literature (Wang et al., 2019), as has a method for accounting for shape information (Achtibat et al., 2023). These need to be adapted and potentially refined for geoscientific data. We look forward to future work in this direction.



**Code and data availability**

The code used the generate the results presented in this paper is available at https://doi.org/10.5281/zenodo.10892412. The ClimateNet dataset (P21) is also publicly available (https://portal.nersc.gov/project/ClimateNet/).


**Competing interests**

The authors declare that they have no conflict of interest.

**Author contributions**

- **Tim Radke:** Main author. Contributed to all sections, lead developer of LRP code. Performed ANN training and relevance analysis.
- **Susanne Fuchs:** Contributed to all sections, in particular to figures. Performed ANN training and subregion-robustness analysis. Jointly worked with TR on all other analyses.
- **Christian Wilms:** Co-supervised TR and SF. Contributed to general discussion and document writing. Contributed to acquiring funding (UHH Ideas and Venture Fund).
- **Iuliia Polkova:** Contributed to general discussion and document editing. Contributed to acquiring funding (UHH Ideas and Venture Fund).
- **Marc Rautenhaus:** Proposed and administrated study. Supervised TR and SF. Central role in discussions and writing of all sections.

**Acknowledgement**

The research leading to these results has been funded (a) by the Deutsche Forschungsgemeinschaft (DFG, German Research Foundation) under Germany's Excellence Strategy – EXC 2037 'CLICCS - Climate, Climatic Change, and Society' – Project Number: 390683824, contribution to the Center for Earth System Research and Sustainability (CEN) of Universität Hamburg (UHH), (b) by DFG within the subproject "C9" of the Transregional Collaborative Research Center SFB / TRR
165 "Waves to Weather" (www.wavestoweather.de), and (c) by the UHH Ideas and Venture Fund. IP acknowledges funding from DFG, Project number 436413914.



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
