# Peer review of "Explaining neural networks for detection of tropical cyclones and atmospheric rivers in gridded atmospheric simulation data"

_Geoscientific Model Development, 2024_

## Author Comment (AC3)

First, we would like to thank both reviewers and the editor for their time spent on reviewing our manuscript and their helpful comments to improve the article. Below, we provide detailed point-by-point replies (blue font colour) to each comment. Citations of reviewer passages are in black italics. Revised/added text is highlighted in yellow. Line numbers refer to the resubmitted manuscript.

**Review 1:**

**Summary**

*The authors adapt the explainable AI technique "Layer-wise Relevance Propagation" (LRP) to the semantic segmentation task of detecting atmospheric rivers and tropical cyclones from atmospheric data using convolutional neural networks. LRP enable authors to assign a relevance score to weight the contribution of each pixel in the image in classifying an area to be of a certain type, assessing whether the pixel contributed towards the classification output, against it, or was irrelevant. The usage of LRP was explored through a case study, an analysis of results over the dataset, and a few examples of applications. The performed analysis was interesting and the research questions were clearly formulated. The demonstrated application areas are relevant to geosciences and clearly add to the value of the manuscript. Despite the analysis being generally satisfying, there some inconsistencies and issues that demand attention before publication.*

We would like to thank Referee #1 for the positive feedback on our work. The provided comments very much helped to further improve the manuscript.

**General comments**

- *Inset plots are confusing to the eye, as one tends not to see the bounding boxes drawn because of the grid lines. Also, coastlines are drawn with the same width in the inset and outer plot, reducing the impression that one is looking at a zoomed panel. Perhaps the plots could be made a bit easier to look at.*

**Answer:**

Thank you very much, we agree these figures could be improved. We increased the line width in the zoomed-in region and added a shadow around the inset so that it is easier to recognize. We replaced all figures with insets in the manuscript and in the supplement.

- *219 time steps leading to 459 total mappings is a very small dataset size for deep learning, that will definitely limit the patterns that are learnable without over-fitting a neural network consisting of millions of parameters. Although this issue has probably already been talked about in preliminary works using the same dataset, I believe it merits at least a short discussion in the context of analyzing the spatial patterns relevant to decisions made.*

**Answer:**

Thank you, this is a very good point. We agree that the manuscript benefits from a short discussion. The dataset indeed is small compared to the size of datasets typically used for classification tasks. However, for semantic segmentation, it also needs to be considered that for each of the 459 mappings in the ClimateNet dataset several thousands of classification tasks for individual grid points are learnt. Nevertheless, we agree that it would be desirable to have a larger dataset at hand (P21 also note this issue in their conclusion). This challenge has also been discussed in the medical image segmentation literature, where studies faced similarly small datasets (e.g., Rueckert and Schnabel, 2020; Avberšek and Repovš, 2022).

Nevertheless, in our setup the IoU scores for the test part of the data as listed in Table 2 of the manuscript are very close to the IoU scores of the training part of the data (not listed in the manuscript, for the CG-Net implementation by KS20, batch size 10, we obtain: AR = 43%, TC = 37%, BG = 95%, AR-TC-BG mean = 59%, for comparison on test data as shown in Table 2: AR = 40%, TC = 36%, BG=94%, AR-TC-BG mean = 57%). If overfitting to the training data was present, we would expect these scores to be different (much higher IoU for the training data than for the test data). Also, the ClimateNet dataset contains differing labels by multiple experts for many time steps. These may be effective in avoiding overfitting. Even if overfitting was present, we would not expect LRP results to be "wrong". Instead, we would expect physically unplausible patterns to show up in the analyses we presented.

To mention this issue in the manuscript, we added the following text to the end of Sect. 3 (LL 182ff):

We note that the size of the ClimateNet dataset (cf. Sect. 2) can be considered small for training a deep CNN, a challenge also encountered, e.g., in the literature for medical image segmentation (e.g., Rueckert and Schnabel, 2020; Avberšek and Repovš, 2022). P21 stated they expect CNN performance to improve if a larger dataset was available. However, we also note that ClimateNet's characteristics of containing differing labels by multiple experts for many time steps may be effective in avoiding overfitting. Also, it may limit achievable IoU scores. If strong overfitting was present, we expect physically unplausible structures to show up in the LRP results.

**References:**

Rueckert, D. and Schnabel, J. A.: Model-Based and Data-Driven Strategies in Medical Image Computing, Proceedings of the IEEE, 108, 110–124, https://doi.org/10.1109/JPROC.2019.2943836, 2020.

Avberšek, L. K. and Repovš, G.: Deep learning in neuroimaging data analysis: Applications, challenges, and solutions, Front. Neuroimaging, 1, https://doi.org/10.3389/fnimg.2022.981642, 2022.

- *Since the authors motivate xAI methods for deep learning with the network learning spurious patterns leading to generalization errors, it would have been interesting to look at a case where a detected area is a complete false positive / false negative in the test set, or if there were none, maybe also how the uncertainty in mappings for a specific area related to the certainty of the network, especially as it was claimed that absolute network output magnitude relate to network certainty (L217-218).*

**Answer:**

Thanks, again a very good point. One challenge here is that for the ClimateNet dataset, it is difficult to define false positives or false negatives. As mentioned above, for some timesteps, labels by multiple experts exist, and they also do not necessarily agree.

However, the case that we discuss in Sect. 5 indeed has been labelled by a single expert only. Also, the western AR in the scene has not been labelled by the expert. The question hence arises whether this AR is a false positive of whether the expert missed the feature.

Since you also commented on whether some relevance information could be lost due to summing the contributions of the grid points (see below, second comment under "specific comments"), we decided to put some further analysis of this AR into the manuscript and Supplement. We provide the full answer below in our reply for the "contributions sum" comment.

- *The U-Net and CG-Net implementations both use batch normalization after convolutional layers, at the very least in most of the modules. This has for effect to negate the effect of the bias in convolutional layers (check Ioffe and Szegedy (2015), p. 5). In addition, the CG-Net implementation doesn't apply biases to any of its convolutional layers. This is inconsistent with the manuscript making it seem like the networks used are heavily reliant on the bias terms of layers. Could the authors provide an explanation and a corrected reasoning for their results?*

**Answer:**

Thank you very much for the comment, we agree that this is an important issue and that our manuscript benefits from further clarification. Indeed, both U-Net and CG-Net implementations use batch normalization layers that follow the convolutional layers. Also, the bias neurons and weights are deactivated. However, their effect is subsumed by the batch normalization layers. In the implementation we used, bias neurons and weights are reintroduced for computation of relevance. We describe this process in the following paragraph that we added to Sect. 4 in L 222ff:

For our setup we note that, in contrast to the ANNs used by M22, CG-Net and U-Net as used in the present study contain batch normalization layers (Ioffe and Szegedy, 2015). These layers apply z-score normalization to the output of the convolutional layers. Additionally, the normalized values are shifted and rescaled according to two learned parameters $\beta$ and $\gamma$. This normalization cancels the bias effect. It is, however, subsumed by $\beta$ and still present. In practical implementations, the bias neurons and weights are hence deactivated during training (Ioffe and Szegedy, 2015). This becomes relevant for the computation of relevance, which can be done separately for convolutional layers (Bach et al., 2015) and batch normalization layers (Hui and Binder, 2019). Alternatively, implementations have been proposed in which both layers are merged, with the advantage that relevance needs to be

computed only for the merged layer (Guillemot et al., 2020). In our work, relevance needs to be computed many times for a given ANN (for grid points and features). For efficiency we choose the second option, as a merged layer needs to be computed only once. The merged layer weights are inferred from the original two layers. In particular, the bias weights are reintroduced, and the ignorant-to-zero-issue persists as in M22.

**References:**

Bach, S., Binder, A., Montavon, G., Klauschen, F., Müller, K.-R., and Samek, W.: On Pixel-Wise Explanations for Non-Linear Classifier Decisions by Layer-Wise Relevance Propagation, PLOS ONE, 10, e0130140, https://doi.org/10.1371/journal.pone.0130140, 2015.

Guillemot, M., Heusele, C., Korichi, R., Schnebert, S., and Chen, L.: Breaking Batch Normalization for better explainability of Deep Neural Networks through Layer-wise Relevance Propagation, 2020.

Hui, L. Y. W. and Binder, A.: BatchNorm Decomposition for Deep Neural Network Interpretation, Cham, Book Title: Advances in Computational IntelligenceDOI: 10.1007/978-3-030-20518-8_24, 280–291, https://doi.org/10.1007/978-3-030-20518-8_24, 2019.

Ioffe, S. and Szegedy, C.: Batch Normalization: Accelerating Deep Network Training by Reducing Internal Covariate Shift, in: Proceedings of the 32nd International Conference on Machine Learning, International Conference on Machine Learning, 448–456, 2015.

**Specific comments**

- *L50: Some clarification would be welcome to what is meant by "correct patterns in data".*

  **Answer:**

  Thank you, this should have been "plausible" instead of "correct" – we already used "plausible" throughout the remainder of the manuscript.

- *L212: Is it a possible problem to sum the contribution of different pixels in a shape when some of these pixels may have relevance scores of the opposite sign for a specific relevance location? Is there a risk for losing information here?*

  **Answer:**

  Thank you very much, this is a very interesting point. We agree that the issue is indeed not very clear from our text. We decided that the manuscript would benefit from some additional insight into how the summed composite maps that we discuss in Sect. 5 can be further decomposed into positive and negative relevance. We considered it valuable to provide a brief analysis of which regions of the western AR in the inset of Fig. 3 in the example contribute positive and negative relevance to the composite. We think this helps the reader to better understand the composite maps.

  As mentioned above, the western AR has also not been labelled by the human expert. We hence use the relevance decomposition to further judge the plausibility of the detection, to investigate whether the AR is a false positive of whether the expert has missed the feature.

We inserted the following text in Sect. 4 following L 237ff, and a discussion and an additional figure in Sect. 5 at line L 359ff:

Sect. 4:

[revised manuscript text omitted]

- *Regarding the selection of most relevant input variables (Application 1), it likely is a problem that there is no separate validation and test set. A more correct experimental setup would have assessed the relevance of each variable on a validation set and assessed the retrained networks on a separate test set. However, it can be understood that this was not done because of the already limited size of the dataset. A few sentences on the potential implications of this on the validity of the results could be added.*

**Answer:**

Thank you very much for the comment. As already mentioned above in our comment regarding the size of the dataset, we see no evidence of overfitting for our setup (as stated

above, IoU scores for training and test data are close to each other for the setup using the TMQ-U850-V850-PSL as inputs; we observe the same when training CG-Net with all 16 inputs). Also, our analysis of the computed relevance in the manuscript showed no signs of physically unplausible patterns. Hence, we argue that using only the test set without a validation set already provides insight into the importance of different variables. We agree, however, that a more correct approach would follow your comment and use a validation set. We added the following sentences at line 552:

We also note that since the size of the ClimateNet dataset is limited (cf. Sects. 2 and 3), we split the data into training and test parts only (cf. Sect. 2). Relevant variables were determined based on the test data (Fig. 10). The retrained CG-Net setups in Table 3 were again evaluated on the test data. Some care needs to be taken with the results of this approach, as the variables found to be relevant could potentially be relevant mostly for the test data. If a larger dataset was available, an improved setup would split the data into three parts, also including a validation part (e.g., Bishop, 1995) for evaluating the results of the retrained setups.


- *Ensure that italics are not used sparingly throughout the text.*

  **Answer:**

  We assume that you mean "use italics only sparingly" (instead of "*not* use")? It is true that we used italics to emphasize parts in sentences where we think that they could be misunderstood otherwise (following the guidelines: "Italic font may be used for emphasis, although this should be used sparingly (e.g. data were *almost* consistent)."). We have, however, revisited the manuscript and reduced the number of occurrences.

- *The abstract effectively summarises the paper but could better highlight the novel contributions. Mentioning specific findings or results in more detail would strengthen the impact, e.g. stating which were the most relevant input variables found. Consider adding some quantitative elements in the concluding sentences to replace general sentences such as "...which helps to build trust in the approach" with a relevant statistic.*

  **Answer:**

Thank you for pointing this out, we agree. We revised the abstract and added some statements from the conclusions:

Detection of atmospheric features in gridded datasets from numerical simulation models is typically done by means of rule-based algorithms. Recently, the feasibility of learning feature detection tasks using supervised learning with convolutional neural networks (CNNs) has been demonstrated. This approach corresponds to semantic segmentation tasks widely investigated in computer vision. However, while in recent studies the performance of CNNs was shown to be comparable to human experts, CNNs are largely treated as a "black box", and it remains unclear whether they learn the features for physically plausible reasons. Here we build on the recently published "ClimateNet" dataset that contains features of tropical cyclones (TCs) and atmospheric rivers (ARs) as detected by human experts. We adapt the explainable artificial intelligence technique "Layer-wise Relevance Propagation" (LRP) to the semantic segmentation task and investigate which input information CNNs with the Context-Guided Network (CG-Net) and U-Net architectures use for feature detection. We find that both CNNs indeed consider plausible patterns in the input fields of atmospheric variables. For instance, relevant patterns include point-shaped extrema in vertically integrated precipitable water (TMQ) and circular wind motion for TCs. For ARs, relevant patterns include elongated bands of high TMQ and eastward winds. Such results help to build trust in the CNN approach. We also demonstrate application of the approach for finding the most relevant input variables (TMQ is found to be most relevant, while surface pressure is rather irrelevant) and evaluating detection robustness when changing the input domain (a CNN trained on global data can also be used for a regional domain but only partially contained features will likely not be detected). However, LRP in its current form cannot explain shape information used by the CNNs, although our findings suggest that the CNNs make use of both input values and the shape of patterns in the input fields. Also, care needs to be taken regarding the normalization of input values, as LRP cannot explain the contribution of bias neurons, accounting for inputs close to zero. These shortcomings need to be addressed by future work to obtain a more complete explanation of CNNs for geoscientific feature detection.

- *The introduction is well structured but somewhat dense. It could be improved by breaking up longer sentences into shorter, more digestible parts for the reader.*

**Answer:**

Thank you for pointing this out, we agree that some sentences have become unnecessarily complex. We revised the entire manuscript (all sections) and broke up long sentences where it seemed appropriate to us.

**Specific comments:**

- *L12 Recently, also the feasibility of learning → Recently, the feasibility of learning feature detection tasks using*

  **Answer:** Thank you for pointing this out, we changed the sentence accordingly.

- *L33 "Features are typically objectively detected based on a set of physical and mathematical rules-"→"Features are typically detected based on a set of physical and mathematical rules"*

  **Answer:** Thank you for pointing this out, we changed the sentence accordingly.

- *L34 For example, cyclones can be identified by means of searching for minima or maxima in variables including mean sea level pressure and lower-tropospheric vorticity → For example, cyclones can be identified by searching for minima or maxima in variables including mean sea level pressure and lower-tropospheric vorticity*

  **Answer:** Thank you for pointing this out, we changed the sentence accordingly.

- *L38 Recent research, however, has shown that given a pre-defined labelled dataset… → Consider "Recent research has shown that, given a pre-defined labelled dataset…"*

  **Answer:** Thank you for pointing this out, we changed the sentence accordingly.

- *Line 50: CNN --> CNNs*

  **Answer:** Thank you for pointing this out – this was intended to be singular, but an "a" was missing. We changed to "a CNN".

- *Line 120: "contained the P21" should be "contained in the P21."*

  **Answer:** Thank you for pointing this out, we changed the sentence accordingly.

- *Line 410: "linearly scales all inputs to the positive range [0..1]" should be "linearly scales all inputs to the range [0, 1]."*

  **Answer:** Thank you for pointing this out, we changed the sentence accordingly.

- *L165 "For instance, a batch size of 10 reduces training time for a single run by 10% while achieving similar evaluation results.": The phrase "achieving similar evaluation results" might be clearer, a suggestion could be "without significantly deviating from the evaluation results."*

  **Answer:** Thank you for pointing this out, we changed the sentence accordingly.

- *Line 195: "In our case, activation $a$ and relevance $R$ are 3-D grids with size of the current layer-dependent horizontal grid times the number of classes" is a confusing sentence. Consider rephrasing for clarity.*

  **Answer:**

  Thank you, we changed the sentence to:

  Note that in our case, activation $a$ and relevance $R$ are 3-D grids. Their size is given by the size of the 2-D data grid of the respective layer, times the number of classes.

- *L203 referred to by M22 as the 'ignorant-to-zero-input issue*

  **Answer:** Thank you for pointing this out, we changed the sentence accordingly.

- *L601 geoscientific science community → geoscientific community*

  **Answer:** Thank you for pointing this out, we changed the sentence accordingly.

**Acknowledgements:**

- *Put this section based on the author contibution from:*

  *https://www.geoscientific-model-development.net/submission.html. Names in acronyms, I.e Jhon Smith → (JS)*

  **Answer:**

  Thank you, we checked the linked submission requirements and changed this section accordingly to:

  ==TR worked on conceptualization, data curation, formal analysis and investigation, developed the LRP code, performed the CNN training, conducted the relevance analysis, and contributed to writing of all sections. SF performed CNN training and subregion-robustness analysis, jointly worked with TR on all other analyses, and contributed to all sections, in particular to figures. CW co-supervised TR and SF, contributed to general discussion and writing, and contributed to acquiring funding (UHH Ideas and Venture Fund). IP contributed to general discussion and document editing and contributed to acquiring funding (UHH Ideas and Venture Fund). MR proposed, conceptualized and administrated study, acquired funding, supervised TR and SF; central role in discussions and writing of all sections.==

**About reference section:**

- *Add at least one reference from 2024*

  **Answer:**

  Thank you for the comment; we agree that this is a good idea given the fast pace in ML literature. Hence, we added two very recent references in line 50 and 58.  Both are review papers; one covering semantic segmentation (Mersha et al., 2024) and the other covering explainable AI methods (Manakitsa et al., 2024).

---

## Author Response (AR2)

We would like to thank the reviewer and the editor for their time spent reviewing our revised manuscript. We also thank you for your valuable suggestion about our answer to the overfitting concern. In the following, citations of reviewer passages are in black italics. Revised/added text is highlighted in yellow. Line numbers refer to the resubmitted manuscript.

Suggestion:

*The newly added text in the manuscript does not fully substantiate their claim: "If overfitting to the training data was present, we would expect these scores to be different (much higher IoU for the training data than for the test data)." Including evidence of score differences would strengthen this argument. I encourage the authors to provide additional details, such as results from the CG-Net implementation by KS20 using a batch size of 10, to support this statement directly in the manuscript. Thank you.*

To strengthen our claim, we deliberately overfitted the CG-Net implementation by KS20 by training the CNN for 100 epochs instead of 20. This leads to a large gap between IoU scores calculated on the training data (AR = 60.0%, TC = 53.3%, BG = 96.7%, AR-TC-BG mean = 70.0) and test data (AR = 37.8%, TC = 32.5%, BG = 94.4%, AR-TC-BG mean = 55.0%). We add the following lines at line 187 into our manuscript to discuss the deliberate overfitting:

Also, strong overfitting typically results in evaluation scores being distinctly better for the training data compared to the test data (e.g., Bishop, 2007). For example, for the CG-Net implementation by KS20, batch size 10, we obtain the following IoU scores for the training data: AR = 43.6%, TC = 37.7%, BG = 95.2%, AR-TC-BG mean = 58.8%. These scores are very close to those listed in Table 2 for the test data, indicating that no overfitting is present. In comparison, if we deliberately overfit CG-Net by training with 100 training-evaluation epochs (instead of 20), we obtain IoU scores of AR = 60.0%, TC = 53.3%, BG = 96.7%, AR-TC-BG mean = 70.0% for the training data and AR = 37.8%, TC = 32.5%, BG = 94.4%, AR-TC-BG mean = 55.0% for the test data.

Reference:

Bishop, C. M.: Pattern recognition and machine learning, 5. (corr. print.)., Springer, New York [u.a.], XX, 738 S. pp., 2007.

As a last point, we want to thank everyone involved in our review process again for your time and effort.

Best Regards,

Tim Radke, on behalf of all authors